# Rearticulating the Conventions of Hajj Storytelling: Second Generation Moroccan-Dutch Female Pilgrims' Multi-Voiced Narratives about the Pilgrimage to Mecca

**Marjo Buitelaar** 

Faculty of Theology & Religious Studies, University of Groningen, 9712 GK Groningen, The Netherlands;
m.w.buitelaar@rug.nl

**Abstract:** This article explores the interplay between content, narrator, and lifeworld in narrative constructions concerning the meanings of pilgrimage to Mecca by studying the hajj stories of second-generation Moroccan-Dutch women. By adopting a 'dialogical approach' to self-storytelling, it is asked how the pilgrimage experiences of these women and the meanings they attribute to them are shaped by different intersecting discursive traditions that inform their daily lives. It is demonstrated that by creative re-articulation and mixing of vocabularies from different discursive traditions to make sense of their hajj experiences, the women contribute to a modern reconfiguration of the genre of hajj accounts. Since gender is the site par excellence where the public debate about the (in)compatibility of being Muslim and being European/Dutch is played out, specific attention will be paid to how the women negotiate conceptions of female Muslim personhood in their stories.

**Keywords:** pilgrimage accounts; hajj; dialogical approach; intersectionality; (female) Muslim personhood; ethical discourses; Moroccan-Dutch Muslims

## 1. Introduction

At the start of my research project on the meaning of the hajj or pilgrimage to Mecca in the lives of Dutch Muslims with migration backgrounds, a friend of mine suggested I meet her mum and her sister, who were preparing for the hajj, a religious obligation that all Muslims who are capable should perform once in their lives.[1] Her 85-year-old widowed mother, Zulikha, was about to embark on her eighth journey to Mecca. Fatima, her 53-year-old sister, would accompany Zulikha on what for Fatima would be her first visit to Mecca.[2] We met at Zulikha's home. As we were chit-chatting over a glass of mint tea, Zulikha divided her attention between attending to her guests and watching Makkah Live, a television channel that offers 24/7 live broadcasting from the Grand Mosque in Mecca. "Look", she said, her face showing anticipatory pleasure as she pointed at the screen where a throng of pilgrims could be seen circumambulating the Ka'ba, the cuboid building covered with a black cloth in the center of the courtyard of the Grand Mosque. "It's already filling up. *Wallah*, [truly, mb] in three weeks' time, it will be so packed that you'll hardly know where to put your feet". Looking slightly concerned, her daughter Fatima remarked: "I wonder what it will be like to be immersed in such a big crowd . . . ".

---

1. I would like to thank Daan Beekers, Martijn de Koning, my fellow department members and colleague Susanne Luther at the Faculty of Theology & Religious Studies of the University of Groningen, as well as the two anonymous reviewers for their most valuable tips and feedback on an earlier version of the article.

2. To protect the privacy of my friend's mother and sister, Zulikha and Fatima are pseudonyms.

"Just exercise *sabr* [patience, mb], my daughter, and put your faith in God", her mother responded briskly. Jumping to the opportunity to hear more about Zulikha's previous hajj experiences, I asked her: "What is it like for you to do the *tawaf* [sevenfold circumambulation of the Ka'ba]?" In response, she gave an elaborate second-person account of the different rites that make up the hajj. Hoping for more personal stories, I tried another question: "How did you feel standing in front of the Ka'ba for the first time?" Hands and eyes turned upward, Zulikha responded: "What one experiences there is beyond words. *Wallah*, it is so overwhelming, you cry and cry and cry. Some people even faint!" After several unsuccessful further attempts to elicit more specific anecdotes about Zulikha's personal hajj experiences, I took recourse to the kind of 'why' question I always tell my students to avoid[3]: "You have been on hajj before. Why do you wish to go again?" Zulikha looked at me with a puzzled face and then replied: "Why ever not?!". Illustrating that our understanding of the words of our interlocutors is always contingent upon our ability to imagine the worlds they are trying to convey (cf. Squire et al. 2008, p. 14), I could not find the words to formulate what to my friend's mother might be a sensible next question.

At the time I blamed my failure to get Zulikha to share her personal experiences of the hajj on my lack of proficiency in the Moroccan-Arabic dialect. Once I started interviewing women of Fatima's generation, however, it dawned on me that there was more to it. Comparing their own take on the pilgrimage to that of the generation above them, many of the younger research participants pointed out that pilgrims of the age-group of their parents focus more strictly on correct performance of the rites, and do not ponder much on the 'why' of hajj practices. In some cases, parents were even said to be reluctant to do so. Although they would return from Mecca full of stories, according to their daughters they tended to be rather inarticulate about the personal meanings of the hajj experience to them. We discerned a similar pattern in our interviews with older pilgrims. Conversely, reflections on personal meanings of experiences abound in the stories of second-generation Moroccan-Dutch women.

In what follows, I will discusses the hajj stories of female pilgrims who were the first children of Moroccan migrants to the Netherlands to grow up in the country. The purpose of the article is to demonstrate how this relatively new category of pilgrims inscribe their own understandings of the pilgrimage to Mecca in the already existing conventions of hajj storytelling. More specifically, in order to study the women's agency in inhabiting and addressing the power structures that regulate their lifeworlds, the situatedness of their hajj stories in a constellation of different normative ideals is analyzed (cf. Fadil 2011).

To this end, I will discuss the interplay between content, narrator, and lifeworld in the women's hajj stories. I will do so by analyzing the dialogues that the women engage in in their storytelling with a multiplicity of voices that inform their daily lives. I will analyze the women's engagement with a modern re-articulation of the genre of hajj accounts by adding new story lines and modifying existent ones. My argument revolves around the ways the narrators creatively rearticulate and mix vocabularies from various discursive traditions to make sense of their hajj experiences in their stories.

The article thus aims to contribute to pilgrimage studies by providing insights in the interplay between pilgrims' individual experiences and collectively shared conventions of storytelling about pilgrimage. A second aim is to contribute to the production of knowledge on Islam in Europe. Contrary to dominant views in both European public debates and in circles of Salafi inspired Muslims who conceive of Islam as a homogeneous and unchanging religion that is distinct from Europe's cultural heritage, the hajj stories discussed here exemplify that Islam is a complex, living tradition that is articulated in multiple ways across different historical and cultural contexts.

---

3   Reasons to avoid 'why' questions in interviewing about personal experiences are that they tend to produce over-reflected intellectualized rationales and may give interlocutors the feeling that they are being examined or called to account for their acts or views, (cf. Kvale 2007, p. 58).

The specific age-group of female pilgrims I focus on ranges from women in their late thirties to those in their early fifties.[4] One reason why I focus on this age-group is that they represent two categories of people whose participation was low in hajj performance until recently: Women and people before old age. A second reason which makes pilgrims of this particular category interesting to study is that the habitus of Muslims who grew up in the Netherlands is informed by a specific constellation of discursive traditions that differs from the configuration of traditions their parents grew up with. I will argue that the expectations, experiences, and the meanings attributed to the pilgrimage by the women whose stories I will discuss are the result of new sensibilities that they have acquired on the basis of how their positions in various cultural contexts intersect. A third reason why I concentrate on the hajj stories of women is that conceptions about female Muslim personhood constitute the site *par excellence* where the public debate about the (in)compatibility of being Muslim and being European or Dutch is played out. Moreover, exactly because the overall integration of these women in Dutch social domains is more thorough than that of their parents, they also find themselves in situations where their Muslimness is interrogated more often, stimulating reflection on their religious heritage (cf. Göle 2017; Jouili 2015).

## 2. Personal Experiences, 'Grand Narratives', and the Negotiation of Meanings

Narratives do not merely give words to experiences, but experiences themselves are shaped by words, more specifically by the meanings these words have acquired in the vocabularies of the discursive traditions available to narrators as they interpret their experiences (cf. Coleman and Elsner 2003, p. 8). In my analysis of the hajj stories produced in the interviews, I therefore adopt a dialogical approach to the narrative construction of the self that builds on Mikhail Bakhtin's work on 'dialogism' and textual 'polyphony' (cf. Buitelaar 2006, 2013; Hermans and Hermans-Konopka 2010; Bell and Gardiner 1998; Bakhtin 1981). With regards to hajj narratives, this means that such accounts are informed by prevailing views within the specific Muslim community the narrator belongs to concerning the experience and meanings of the pilgrimage to Mecca. More specifically, because Mecca is the holiest city in Islam and hajj performance conceived of by Muslims as literally 'stepping in the footsteps' of the prophet Muhammad and other role models in Islamic historiography, in their hajj stories pilgrims position themselves in relation to specific cultural ideals about Muslim personhood.

In line with Martyn Smith's argument concerning idealized collective stories about sacred places, (cf. Smith 2008, p. 26), the snapshot in the introduction of the conversation between Zulikha, Fatima, and myself illustrates that the accounts of individual pilgrims about their journey to Mecca resonate with a collectively shared and culturally embedded 'grand narrative' about the hajj and convey moral lessons about Muslim virtues. For example, hearing her mother describe the experience of sighting the Ka'ba for the first time as being 'beyond words' and causing pilgrims to cry and sometimes even

---

[4]    A total of 76 interviews were conducted for the hajj research as a whole, 50 of whom were pilgrims of Moroccan backgrounds, of whom 32 were women and 18 were men. Seven female and 3 male Moroccan participants belong to the generation of economic migrants that came to the Netherlands in the 1960s and 1970s. The bulk of the interviews with descendants of migrants was conducted in Dutch by the author of this article, while the interviews with Moroccan older pilgrims were conducted by a research assistant in Tamazight or *darija* (Moroccan-Arabic). She and a Moroccan-Dutch postdoc who also accompanied a group of pilgrims on the hajj journey each also interviewed a few younger female pilgrims. All interviews were audio-taped and fully transcribed. The participants were recruited through 'snow balling', initial contacts in existing personal networks, mosques, and community centers serving as starting points to approach subsequent interviewees. In the first part of the interview, our interlocutors were asked to draw their 'life line' and mark what they considered important phases of their lives so far. The 'life line' was then used to zoom in on specific key events, significant others, achievements, and challenges. This open part of the interview was concluded by reflecting on the interviewee's religious upbringing and personal religious development. Asking for childhood recollections about people who had gone on pilgrimage to Mecca then created a bridge to the second, slightly more structured topical part of the interview, which concentrated on the interviewee's own pilgrimage to Mecca, either in the form of *umra*, the voluntary 'smaller' pilgrimage to Mecca that can be performed any time of the year and is restricted to rites within the confines of the Grand Mosque in Mecca, or the hajj, the compulsory pilgrimage that capable Muslims should perform once in their lives, which takes place between the eighth and thirteenth day of the Dhu al-Hijja, the last month of the Islamic calendar. For this article, I will concentrate on stories about the hajj.

faint, Fatima learned to expect to be emotionally overwhelmed when it would be her turn to stand in front of what to Muslims represents God's house on earth. Also, stating that her daughter should exert *sabr* and put her faith in God in response to Fatima's meditation on the immense crowd circling the Ka'ba, Zulikha conveyed the moral lesson that the hajj is a trial that must be faced with patience and perseverance and an unwavering trust in God. Finally, by answering my question why she wished to visit Mecca again with her rhetorical counter question "Why ever not?!", Zulikha communicated the incomprehensibility if not inappropriateness in her eyes of the suggestion that a Muslim might not desire to go there whenever the opportunity arises.

The conversation between Zulikha, Fatima, and me thus illustrates that expectations of prospective pilgrims are informed by the heritage of hajj stories of those who preceded them, stories that together shape a more or less normative 'grand narrative' about the pilgrimage. Moreover, such stories direct the perceptions of pilgrims as they perform the pilgrimage and provide them with a vocabulary and already existing story lines or 'scripts' to interpret their own pilgrimage experiences. Storytelling is therefore not a simple matter of creating personal meanings, but an intersubjective endeavor, involving a 'politics of experience' in which a multiplicity of private and public interests is in play (Jackson 2006, p. 11). Storytelling, then, is always dialogical; it is a process in which narrator and audience are engaged in an ongoing negotiation of meanings. In this sense, the women who participated in the research are not the sole author of their stories; their hajj accounts are necessarily co-authored and multi-voiced (cf. Zock 2013).

Religious 'grand narratives', however, do not constitute the only genre that informs the personal desires and senses of selfhood of Muslims (cf. Schielke 2009). Besides being informed by the Islamic tradition as transmitted to them by their parents, the habitus and the sense of self of women who grew up in the Netherlands are also shaped by the modern liberal discourse and the culture of consumerism that dominate in the Dutch cultural contexts in which they were educated, earn their living, and lead their social lives. For most Moroccan-Dutch citizens of the migrant generation to which Zulikha belongs, pilgrimage to Mecca is the only alternative journey besides visiting the country of origin that they are familiar with. The horizon of their offspring, however, is much wider. Besides being accustomed with the travel practices of their parents, the personal longings of Moroccan-Dutch citizens of Fatima's generation have also been shaped by growing up in a country where making a holiday trip to 'chill' or explore hitherto unknown territory is almost considered a basic human need. As a consequence, they have expanded their views on desirable travel destinations, and they have high expectations regarding the efficiency and quality of transportation, accommodation, and time-management during their journeys. Also, whereas their parents mostly have rural backgrounds and enjoyed little or no formal education, most interviewees who participated in the research project are higher educated urbanites. As a result, like other modern-educated middle-class citizens in the Netherlands and elsewhere—including many Muslim majority countries[5]—they have incorporated norms about hygiene and punctuality, and liberal values like individualism, gender equality, and self-enhancement.

Moreover, as several anthropologists have demonstrated, the religious trajectories of Muslims who grow up in Europe are at least partially constructed in response to often Islam-hostile public debates about the position of Islam in Europe (cf. Jouili 2015; Fadil 2011, 2008; Buitelaar and Stock 2010; de Koning 2008). The embodied dispositions that form the matrix for the perceptions, appreciations, and actions of European Muslims therefore reflect how a specific constellation of various discursive traditions have simultaneously shaped their 'sensibilities'; the moral and aesthetic dimensions of their experiences and emotional lives. In terms of Bakhtinian psychology, this means that by appropriating words, meanings, and story lines from conventional hajj accounts in their own stories, they intone them and place them in relation to conceptual patterns and values from other discursive traditions, thus reshaping them as they use them (cf. Shotter and Billig 1998, p. 24).

---

5	See, for example (Hafez 2011).

The descendants of Moroccan migrants have therefore become active co-authors of the prevailing 'grand narrative' of pilgrimage to Mecca in their Muslim community and contribute to its further development, inscribing their own understandings of (female) Muslim personhood into existing representations. Hajj performance of younger Muslims is a fairly recent development. As is the case in many other Muslim communities, the participation of Moroccan women is an even greater novelty. The storytelling of second-generation Moroccan-Dutch female pilgrims is therefore both an act of 'emplotment', selecting events and putting them in a specific sequence, and an act of 'emplacement', locating oneself in the existent 'grand narrative' about the hajj (cf. Jackson 2006, p. 31).

There are, however, restrictions on the freedom of narrators to improvise upon and add new dimensions to existing meanings. Much as we can bend conventional meanings attributed to words in established views, if our stories are to be understood, they must be oriented toward the specific conceptual horizon of our listeners. In order to be recognized, the self-presentations of narrators therefore depend on prevailing shared imaginative conceptions of, for instance, a specific kind of personhood. In this sense, storytelling is always informed by the existential tension between 'being for oneself' and 'being for another' (Jackson 2006, p. 30). The freedom of narrators to shape their own stories is therefore far from absolute (Olson and Shopes 1991, p. 193). It hinges on the specific constellation of power operations through which the articulation of certain imaginaries of personhood may be enabled or disabled (cf. Fricker 2007, p. 14; Ochs and Capps 1996, pp. 32–35).

As the listeners to narratives are, in a sense, co-constructors of the stories told, for the specific interview situation in which the hajj stories that are discussed in this article were produced, it is relevant to reflect on who besides the interviewer as primary recipient were the imagined audiences that the interviewed women addressed in their hajj stories. Besides myself as principle investigator, two other female research team members participated in the interviewing, both of the same age-group as the interviewees, and like them having Moroccan backgrounds. This background is likely to have informed the assumptions of my colleagues' interlocutors concerning their familiarity with hajj practices and the conventions of established hajj stories. Both interviewers were trained to respond to the kind of "You-know-what-I-mean" remarks with follow-up questions concerning what the topic discussed meant to the narrator herself. I myself am a woman of non-Muslim Dutch descent in her early sixties, thus belonging to a category of Dutch citizens whom our interlocutors consider likely to have one-sided preconceptions about Islam. In most cases my long-term research in Morocco and knowledge on Islam proved to be helpful in developing rapport. However, living in a context where Muslims are routinely questioned about their religious stance by non-Muslim fellow citizens must have informed how my presence mediated an imagined non-Muslim audience to the women I interviewed. A second imagined audience involved in the dialogical construction of the hajj stories in the interview setting consists of Moroccans and a wider category of citizens with Muslim backgrounds. How these different audiences were addressed varied between different interviewees, but also between different stories in the same interview. As I will argue in the next section, however, in their storytelling the women's efforts were predominantly aimed at connecting both audiences by a narrative strategy in which an Islamic and a modern liberal discourse converge.

## 3. Narrative Convergence of Islamic and Modern Liberal Discourses

Most of our interlocutors positioned themselves specifically as Dutch Muslims with Moroccan backgrounds in their hajj stories.[6] They did this predominantly by comparing their own views and practices to those of pilgrims of the generation of their migrant parents. One woman, for example, stated:

---

6　　Note that I use 'we' and 'our' interlocutors to indicate that while the bulk of the interviews were conducted by myself, some women whose stories are discussed here were interviewed by one of the Moroccan-Dutch interviewers who participated in the research. The singular 'I' is used when presenting my own analysis and argumentation as the sole author of this article.

> Old pilgrims are taking leave of *al-dunya*, [the worldly, mb] and prepare to meet their Creator. Settling debts with God and seeking His forgiveness for their sins is their biggest concern.

Another women explained that:

> Their focus is very much on the cleansing of sins and getting *hasanat* [the credits for good deeds that will be weighed against sins on the Day of Judgment, mb].[7]

In line with the scriptural commandments (cf. Qur'an 2:196–98), fulfilling one's duty to God and seeking forgiveness inform the motivations of nearly all pilgrims. In the hajj practices of older pilgrims, this often translates into a preoccupation with correctly following the rules, such as performing the proper bodily movements and saying the right *du'a*s or supplication prayers at the stipulated time and place, lest one's hajj should not be *maqbul*, accepted by God. "They're very much concerned with ticking the boxes" one woman said about elderly female pilgrims. "They're watching each other all the time and keep warning not to make mistakes", another stated.

While our interlocutors' descriptions of the hajj practices of older pilgrims suggest a rather superficial, ritualistic approach lacking personal reflection, it should be noted that until the late nineteenth century, personal reflections in general were as good as absent in written hajj accounts (Smith 2008, p. 138; McDonnell 1990; Metcalf 1990).[8] A narrative focus on symbolic and personal meanings of the hajj first occurred in pilgrimage accounts of modern educated Muslims at the onset of Islamic reformism in the late nineteenth and early twentieth century (cf. van Leeuwen 2015). The stories of our interlocutors mostly resonate with this reformist approach. When asked what had motivated them to perform the hajj at the particular moment in their life that they did, many younger women obviously first pointed to the obligatory nature of the hajj. They would then be quick to add, however, that other factors had been more important than just duty and the expiation of sins. Again in comparison to pilgrims of the older generation, an interviewee in her forties, for example, explained:

> Older people see it as a duty, as in: "Approaching the end of my life, I *must* do it". A sort of closure so to speak. But for me, it is the beginning of my life.

Additionally, while the accounts of our interlocutors often include descriptions of specific rites, these are not necessarily presented in the chronological order of the ritual hajj 'script'; some are touched upon, while others are not. Moreover, as will be illustrated later in the article, where such descriptions occur narrators tend to dwell on their personal understandings of the meanings of the rites and how the rites affected them emotionally.

Indeed, the hajj stories of many younger women have the character of a narrative exploration of the self. This comes to the fore, for instance, in the often expressed view that contrary to the traditional pattern among older generations to postpone the hajj until 'one is ready' in old age, it makes more sense to perform the pilgrimage early in life. One reason that was mentioned is that hajj performance is physically very demanding, so that being in a good condition helps one to get out most of the experience. What is more, if performed at a young age, the benefits of the spiritual rewards of the hajj can hopefully be reaped for a long time to come. Some women, for example, explained that they had decided to perform the pilgrimage early in their marriage or even for a honeymoon hoping that going through the experience together with their partner would strengthen their marital bond (cf. Kadrouch-Outmany and Buitelaar 2020). Other frequently mentioned reasons were: "Wanting to get back in touch with God and myself"; "Needing a time-out for a spiritual recalibration"; and "Finding a new balance". In line with such motivations, the question of how our interlocutors would summarize

---

[7] Strictly speaking, *hasanat* refer to the 'good deeds' themselves, while the term *ajr* refers to the 'credit points' that can be gained by performing *hasanat*. In practice, the two terms are often used interchangeably.

[8] Also see (Williams 2003) and (Petsalis-Diomidis 2003), who point to similar differences between more 'factual' accounts and personal testimonies in the accounts of Christian pilgrims from different historical and cultural contexts.

the pilgrimage experience elicited responses like: "Personal development"; "A lesson about what really matters in life"; "A spiritual boost"; Tranquillity".

Note that these (desired) beneficial effects hint at hajj performance as a strategy to deal with the effects of today's accelerated culture on the women's everyday lives.[9] Remarkable in this respect was the statement of a woman in her late thirties that she had hoped the pilgrimage would bring her moments of being "totally zen".[10] Linking the aspired effect of the hajj to a popularized notion of the Buddhist concept of 'zen' exemplifies how the stories of the women are not only informed by the Islamic discursive tradition, but equally in dialogue with voices from the wider societal networks in which they operate in the Netherlands. Although the women's hajj narratives concern experiences that are located in the sacred space of Mecca, as a form of 'situated thinking' their storytelling is at least as much oriented towards their everyday lifeworld and encompasses a plurality of perspectives that help them gain an enlarged view of their personal experiences (cf. Jackson 2006, p. 252).

Besides resonating with an Islamic reformist discourse, the hajj stories of our interlocutors are also strongly informed by a conception of personhood that marks the 'subjective turn' in modern culture; the conception of an 'authentic self' who, if choosing to live a religious life, does so because it 'speaks to them' (Taylor 1991, 2002). The ideal of an 'authentic self' implies an understanding of life according to which every individual should realize their own way of life through self-exploration and self-expression rather than "surrendering to conformity with a model imposed from the outside, by society, or the previous generation, or religious or political authority" (Taylor 2002, p. 83). It concerns a shift away from a 'life as' (mother, wife, etc.) governed by fulfilling external roles, duties, and obligations, and a turn towards a 'subjective life' lived by reference to one's own subjective experiences (Heelas and Woodhead 2005, p. 2).

Although the discourse of the 'authentic self' suggests a liberation from conventional social constraints, it also dovetails with the demands of today's global neoliberal political economy. It carries in it the normative imperative to take full responsibility for one's own life and wellbeing.[11] The booming consumer market for products appealing to feelings, wellness, and fulfilment, for instance, taps into the needs of individuals to recuperate and keep up with the accelerated pace of their lives in modern culture.

The desire for "finding a new balance, "personal development", or "a spiritual boost" through hajj performance illustrates that most women have incorporated elements from the subjective-life stance in their narratives. Note, for instance, how one woman expressed having chosen for Islam, being inspired and reinvigorated by the hajj, and conceiving of the pilgrimage experience as something that can help her carry on:

> It [the hajj, mb] is the moment that inspires and reinvigorates you and your *iman* [faith, mb] so that you can carry on. I prefer doing that halfway my life rather than at the end of it. The idea is to get a taste of ["opsnuiven", mb] why you have chosen for this religion.

In fact, hajj tour operators and commercial Muslim platforms appeal to these modern desires to attract new clients, as the following advertisement on Mvslim.com illustrates: [12]

> It's the end of the year [2019, mb], and while we're heading towards a new decennium, some major soul searching questions come up. We reflect about our past, the choices we've made, and, more importantly, the persons we would like to be in the future. Within our highly

---

[9]　See Beekers (2018, p. 82), who found that for both young Dutch Muslims and Christians, performing prayers in a similar way helps them to decelerate (and fit in God in their hectic lives).

[10]　The statement was made by a woman who was interviewed by one of the two Moroccan-Dutch interviewers. She was thus not 'translating' her vocabulary to a non-Muslim Dutch interlocutor.

[11]　It should be noted that the liberation from power structures promised by a discourse of contemporary spirituality that accompanies the 'subjective turn' does not always bring the personal freedom or gender equality that spiritual practitioners seek, but tend to create their own power structures (cf. Fedele and Knibbe 2013).

[12]　https://mvslim.com/new-year-new-me-what-if-you-could-go-to-hajj-for-free/, most recently accessed 1 June 2020.

competitive societies, it sometimes feels as if we don't have a grip on our own lives, we constantly worry about the never ending deadlines and expectations. That's why we search for ways to escape the pressure that we feel. Throughout the history of mankind, people used spiritualism to find inner peace [ . . . ] This spiritual journey takes up a central place within the lives of Muslims. Realising that one of the most important spiritual journeys of Muslims, the Hajj, is too expensive for most of us, Manzil now offers its first ever Hajj-giveaway.

In keeping with the suggestion of horizontally organized social relations in the cultural discourse of the 'authentic self', the understanding of the meanings of the hajj coming to the fore in the stories of our interlocutors tend to depart from a conception of the divine as a loving, nurturing God with whom one entertains a close, personal relationship. This god-image diverges significantly from the representation imparted on the women by their migrant parents of the panoptic surveillance of a God who enforces obedience through punishment.[13] One woman, for example, compared her own conception of God to the one her parents had brought her up with as follows:

> They spoke predominantly in terms of fear for Allah, not in terms of love for God. It was: "If you don't pray you go to hell." rather than: "You should believe because that makes you happy." [ . . . ] I apply a radically different approach with my own children. Until they're teenagers we won't even mention hell or other negative things. In my view, if you learn how to do things out of love [for God, mb], then it's much easier to keep doing them.

I would argue that developing a conception of a loving God is a response to the authoritarian pedagogical parental style that the women have come to question as a result of being equally socialized in a Dutch educational and wider environment governed by a more authoritative, liberal pedagogy (cf. Pels et al. 2006). Furthermore, viewing God as a loving power also opposes the conception of a vengeful God that dominates in more radical understandings of the Islamic tradition that have gained ground in the Netherlands, particularly under Muslims of younger generations. It is particularly this radical strand of Islam that dominates in negative views on Islam among Dutch non-Muslims. In this respect, conceiving of God as a loving power can be seen as challenging various exclusivist discourses in Dutch society at the same time. Most important for my argument here is that the image of God as a power whom you choose to worship not because you fear divine punishment, but because it makes you happy, points to the convergence of an 'ethics of authenticity' with an 'ethics of submission' that focuses on the deliberate pursuit of pious self-cultivation through practices and discourses of submission to a transcendent God (Mahmood 2005).[14]

The meanings our interlocutors attribute to the hajj in their stories reflect this conception of a loving God. For instance, wishing to perform hajj to 'treat oneself' or needing a 'spiritual boost', is very different from a motivation based on strict obedience to God's commands. Additionally, although most women strive to carry out the rites correctly, many of them consider it of no grave consequence to make a mistake. In their view, for God it is one's '*niya*' (intention) that counts first and foremost. Similarly, while they highly value the notion of the expiation of sins, the women see hajj performance less as a radical break from a former life, as most older pilgrims do; nobody's perfect, so while striving to do good, one is bound to make mistakes again upon return from Mecca, and these will not count double,

---

[13]  Also see Beekers (2015, p. 145) and Jouili (2015, pp. 64–68) who report a similar representation of a loving God among young European Muslims.

[14]  For similar reflections on the convergence among these seemingly contradictory ideals in religious expressions of contemporary young European Muslims see: (Beekers 2015; Jouili 2015; Fadil 2008; Roy 2004).

as many older people believe. [15] From a Foucauldian perspective, one could summarize the meaning of the hajj for our interlocutors as a 'technique of the self' that helps keep improving on themselves.

One woman, for example, reflected on the symbolic meaning in her own life of the *sa'y*, the rite of 'running' between the hillocks of Safa and Marwa in commemoration of the search for water by Hajar for her son Ismael after Ibrahim had deserted them in the desert. In our interlocutor's understanding, Hajar's story conveys a lesson about personal empowerment and accountability:[16]

> Like Hajar you may feel that you've been left alone by others. But she trusted in God. It's a story about the kind of ordeals we all face in our lives, the kind of "After rains comes sunshine [in Dutch: "Na regen komt zonneschijn", mb] in terms of the Dutch expression. Hajar did not passively yield to the circumstances or blame Ibrahim. Nor did she wait for God. She took action herself, just as the situation demanded.

Note the ease with which this research participant cross-references the meaning of the *sa'y* to her to the Dutch proverb "After rain comes sunshine". Besides myself as the first recipient of her story, she thus specifically addresses an imagined Dutch speaking audience. By equating the moral lesson she draws from Hajar's story with that conveyed by the Dutch proverb, she positions herself as a Dutch Muslim and challenges commonly held views about the incompatibility of Islam and Western values.

Equally interesting in relation to the impact of different discursive traditions on her conception of self, is the emphasis on agency and self-accountability in her understanding of the *sa'y*.[17] Most contemporary commentaries by Muslim scholars on the hajj prioritize the stories of Ibrahim and Muhammad over those relating of Hajar (Bianchi 2004, p. 29). An exception is the modernist thinker Ali Shari'ati, who presents Hajar as a model of hope and of trust in God (Shari'ati 2014, p. 47). In addition to hope and trust, the woman quoted above mentions connotations that are very much in line with the modern liberal value of taking one's life into one's own hands.[18] Moreover, she attributes meanings to the story of Hajar that transcend Islamic particularities and point to what to her is a universal value: Personal agency. Illustrating the dialogical nature of storytelling, she appropriates elements from the Islamic 'grand narrative' about the hajj and mixes them with the vocabulary from the discourse about the authentic' self, thus producing a multi-voiced hajj account.

Connecting 'Islamic' and 'Dutch' or 'Western' culture by pointing to universal moral lessons that can be learned from the hajj also occurred in other interviews, particularly by interlocutors who are engaged as community volunteers or work in public services.[19] Often, for example, women would point to the hajj as an exemplar of, or exercise in global solidarity and a celebration of diversity. For this reason, some expressed regret that Mecca is forbidden territory for non-Muslims, a regulation that, in their eyes, goes against the spirit of Islam.[20] Drawing on a Muslim perspective for a wider engagement with other people regardless their faith, culture, and lifeworld resembles the stance of 'Islamic cosmopolitanism' that Morris (2019) reflects on in his discussion about the societal engagement in circles of young Muslims in the West. Contrary to the trend of 'deterritorialized' or post-national Islamic discourses discussed by Roy (2004), Morris draws the attention to a more 'rooted' Islamic cosmopolitan discourse among young European Muslims that can also be recognized in many of the hajj

---

[15] To be sure, fear for God's punishment was not completely absent in the stories. A few women, for instance, indicated that it continued to gnaw at them that they had failed to execute a specific rite correctly and that they were torn between feeling reassured that God is forgiving and judges one by one's intentions, and the fear that their hajj might not be accepted after all. While longing to visit Mecca again was a recurring theme in most interviews, for these women repairing previous failure was their primary motivation to do so.

[16] Hajar = Hagar, Ibrahim = Abraham in the Judeo-Christian traditions.

[17] For an explicitly 'feminist' interpretation of the *sa'y* focusing on Hajar as a single mother, see Nomani (2005), whose hajj memoir I discuss in (Buitelaar 2020).

[18] It must be noted that the expression 'help yourself and God will help you' also exists in the Arabic-Islamic tradition, but the 'may God help you' is used much more commonly among Moroccans.

[19] See for example, the hajj story of Farida that I discuss in (Buitelaar 2013).

[20] Some were quick to add that they could understand that you would not want tourists to walk in the way during the hajj season itself, when Mecca comes apart at the seams.

stories. While having a radial outlook, such cosmopolitan engagement is 'rooted' in local experiences and identities. Following Morris, I would argue that exactly because they are often expected to defend their national loyalty on the one hand, while being confronted with exclusivist discourses on Western citizenship on the other hand, articulation of a more encompassing Islam-inspired framework of universal human rights, for example by taking the hajj as a symbol of diversity, reconfirms the women's sense of simultaneous belongings to Dutch society and to the *umma*, the global community of Muslims (cf. Morris 2019, pp. 31–32).

Like the woman quoted above, several other interviewees also explicitly inserted their own 'Dutchness' in their understanding of the pilgrimage to Mecca. A woman in her early forties, for example, stated that it had annoyed her father that she kept asking why certain rites should be performed: "You and your Dutch why questions! Don't ask why.", she quoted him saying, and then added: "That is that Dutch part of me. It can really drive my parents crazy". Whether they realized or not, this father–daughter conversation touches upon a fundamental debate in classical Islamic theology about the purpose and meaning of the hajj. In a particularly elucidating article, Katz (2004) expounds on the different views that Muslim thinkers in different historical and cultural contexts have formulated on the meanings of the hajj and its ritual efficacy. Most build on the influential Muslim scholar Al-Ghazali (d. 1111 C.E.), who formulated a thesis about the pilgrimage to Mecca as a necessarily 'meaningless' ritual. In the view of Al-Ghazali, hajj performance should be an act of unswerving devotion to God's command, and should not be diluted by wishing to understand it. At the same time however, Al-Ghazali did allow for the merits of 'contemplation about salutary moral examples' (Katz 2004, p. 114).

It falls beyond this research-project to study how the views and practices of the women interviewed might be informed by textual sources concerning historical theological debates.[21] Note, however, that the "Don't-ask-why" admonition quoted above is very much in line with Al-Ghazali's view that one should not seek to understand the hajj, as its performance is ideally an act of pure submission.[22] Illustrating the complex and shifting relation between the concepts of 'sincerity' and 'authenticity' in different cultural contexts, submitting to God without questioning points to a different conception of authenticity than the modality that features in the discourse of the authentic self (cf. Sjørslev 2013; Trilling 1972). For Muslims whose self-conceptions have been informed by the discourse of authenticity according to which being religious should be your own choice because it speaks to you, obedience alone is not enough; they must also have the feeling that they get something out of it in their present lives. Indeed, as we will see in the next section, nearly all hajj narratives contained episodes in which our interlocutors had either difficulties in translating one discursive tradition into another or simply refused to do so.

## 4. The Predicament of Voicing Dissidence

A recurring topic in the interviews with pilgrims who grew up in the Netherlands is a critique on the hajj management by the Saudi regime. Notorious in this respect is the lack of hygiene and privacy in the tent camp in Mina, an area five kilometers outside Mecca from where several rites of the hajj are carried out. Pilgrims spend three nights in Mina, where they are accommodated in sex-segregated tents. Provisions in the tent camp are very basic. Being accustomed to having ample personal space and a considerable standard of hygiene in their daily lives means that the lack thereof in the tents, where one lies shoulder to shoulder with eighty to a hundred fellow female pilgrims, came to many of our interlocutors as a "culture shock", as several women put it. Describing the facilities in Mina, some

---

[21]  See (Groeninck 2017); (Beekers 2015); (Jouili 2015); (Fadil 2008), and (de Koning 2008) who studied organized Islamic knowledge acquisition among young European Muslims.

[22]  This attitude might well also explain why Zulikha, whose "Why ever not?" I quoted in the introduction, answered my question why she would want to visit Mecca once again thus: Obligatory or not, if one conceives of hajj performance as an act of 'unswerving devotion', indeed why-ever-not go again and again if given the chance?

interlocutors vented their irritation about the "rip-off" or "lack of value for money" and complained that the circumstances in the tent camp had hampered their spiritual engagement. It angered them that tour guides would respond to their complaints by referring to the Islamic virtue of exercising *sabr*, patience. As assertive middle class urbanites, these women were not willing to expand the religious virtue of patience to encompass acceptance of substandard arrangements.[23] Others avowed that although they had disliked their stay in Mina very much, they considered it a valuable lesson in what you can learn from being forced to leave your own comfort zone.

Another uncomfortable part of the hajj is the night that pilgrims spend in the open air in Muzdalifa. Several interlocutors confessed that having imagined a serene night in the desert under a canopy of stars, they were disillusioned to find they had to struggle to find a place where to put their sleeping mat among millions of other pilgrims, and that it was impossible to avoid the stench emanating from the toilet blocks. For some, this was a lesson in humility, a reminder that we are born empty-handed, and we return to God empty-handed. For others, the experience reminded them of the dire circumstances in Syrian refugee camps and urged them to include the needy in their prayers. For yet others, like the woman quoted below, the experience was almost too much to handle and beyond comprehension:

> Muzdalifa was one of the most intense ["heftige", mb] moments in my life. When we arrived there, I thought: 'What am I doing here? Let's get out of here.' We had to work our way through the crowd to get a place to sit down. No tents, no nothing, all you see around you are millions of people. And right there in the middle of that huge crowd someone was baking French fries [ . . . ] I know Muzdalifa is an important part of the hajj, but I couldn't see the beauty or purpose of it all. You know, that moment should be so enjoyable and zen, but I was baffled and repulsed. To me, Islam is logical. Not seeing the rationale behind it, I thought: 'Something is just not right here'. At the same time, I told myself: 'I should not think so much.' You know, my parents afterwards said: 'What is it that you would have liked to hear about it [before going on hajj, mb]?'

Besides the use of the word "zen" that was discussed earlier in this article, two elements in this excerpt are relevant for my argument here. The first concerns the inner dialogue through which the narrator is trying to come to terms with her ambivalent feelings about her experiences in Muzdalifa. On the one hand, she finds it difficult to submit herself to a rite that she cannot see the meaning of, a line of reasoning representing the voice of the ethical model of the 'authentic self', according to which religion should speak to one. On the other hand, there is the internalized voice of her parents who have taught her to "not think so much", thus representing an ethical model of unreflective obedience to God's commands.

Secondly, the narrator quotes her parents as posing the rhetorical question "What is it that you would have liked to hear about it?" Although she herself does not answer the question in her story, I can almost hear most other interviewees of her age-group exclaim in unison: "The truth". The truth, that is, about the actual circumstances during the hajj—including the ugly bits—rather than the ideal picture pilgrims of the generation of their parents tend to sketch. Many younger female pilgrims who had relied on the one-sided stories of their parents stated that if only they had known what to expect, they would have come better prepared and not been thrown off their guard as much as they had been. Most older Moroccan pilgrims, however, find it disrespectful if not blasphemous to include negative stories in their hajj accounts; one should be grateful for having been "called" to Mecca by God, and not speak bad about the holiest city of Islam that hosts God's house on earth.

Like older pilgrims, most of our younger interlocutors feel that being put to the test is part of the hajj performance. Unlike many older pilgrims, however, they distinguish between the demands

---

[23] See (Haq and Jackson 2009), who noted similar 'demanding' attitudes among younger Pakistani-Australian pilgrims. Also see (Kadrouch-Outmany and Buitelaar 2020), who discuss women's views on the unequal gendered division of space in Mina and other hajj.

of religion on the one hand, and the politics and management of the pilgrimage on the other.[24] The first should be respected, the second can be criticized. The two intertwine, however, in Saudi policies concerning sex-segregation. While the actual hajj rites such as the *tawaf* and *sa'y* are mixed, for the *salat*, the five daily prayers, as well as for the Friday sermon, women are relegated to the back of the courtyard of the Grand Mosque in Mecca, which prevents them from having a good view on the Ka'ba. Similarly, the female section of the *rawda*, the part of the mosque in Medina where the prophet Muhammad is buried, is significantly smaller than the male section, and screens block the shrine enclosing the tomb from women's view. Additionally, the opening times for women to enter the *rawda* are considerably shorter.[25]

Many of our interlocutors expressed indignation about the impact of such institutional gender-discrimination on their spiritual experience. Notably fewer women volunteered information about gender discrimination by fellow pilgrims. Fatima, who had agreed to be interviewed after her return from Mecca when I had met her in the home of her mother Zulikha, was exceptional in her willingness to share an experience of physical harassment. Having accidentally been pushed by the person walking next to her during the *tawaf*, she had bumped into a male pilgrim. In response, the man had punched her hard in the ribs with his elbow while loudly exclaiming to the man next to him: "What are these women doing here crowding this place anyhow?" The incident was beyond anything Fatima had imagined possible to happen during the hajj. She was shattered. Her chest hurt for several weeks. What hurt her even more was the realization that particularly older women have been willing to cover up abuse and other instances of gender discrimination for religious reasons, thus contributing to the perpetuation of their own subordination:

> This is not the kind of things you easily share. When you get back and people ask you about your hajj, you say: "*Allah la ykhaty shi wahed min dhalikal makan*". Something like: "Hopefully God will take everybody to that place". [ . . . ] Mind you, I DO tell women what to expect, so that they will be prepared. But women like my mum say: "*ster ma star Allah*": cover up what Allah covered up . . . you know, no airing dirty laundry. They say it brings bad luck.

Speaking out about gender discrimination by fellow pilgrims puts Fatima in a predicament. Besides feelings of unease about violating the norm about not "airing dirty laundry", she runs the risk of being accused by fellow Muslims for selling out to an already prejudiced non-Muslim audience that might read her stories as proof of what they consider the intrinsic misogynist nature of Islam. In relation to both Muslim and non-Muslim listeners of her story, then, Fatima has to grapple with 'testimonial injustice'; the injustice of her experiences not being 'heard' or recognized by the very groups in society she identifies with (cf. Fricker 2007).

Fatima's decision to tell other women about her experiences reminded me of the Pakistani female pilgrim who in 2018 shared her experiences of sexual assault during the *tawaf* on Facebook. Her post was quickly deleted, but not before being picked up by the Egyptian journalist Mona Eltahawy. Eltahawy had written about a similar experience previously, but her story had practically gone 'unnoticed' (Eltahawy 2015, pp. 48–53) In the wake of the #MeToo movement, however, this time it was picked up, and soon other women started to share similar experiences under the hashtag #MosqueMeToo. Besides receiving statements of support, the movement was also criticized on social media as being a tool of Western anti-Islam propaganda.[26] A google search for the hashtag reveals that most blogs and tweets were posted in 2018, suggesting that the movement has lost much of its momentum in the past two years.

---

24　Also see (Al-Ajarma 2020) who discusses the views of pilgrims from Morocco on Saudi Hajj politics.
25　While visiting Medina is not part of the hajj itself, most tour operators include a visit in Medina so that pilgrims can pay their respects to the prophet Muhammad.
26　Cf. https://www.bbc.com/news/world-43006952; https://crcc.usc.edu/sexual-assault-during-hajj-will-mosquemetoo-lead-to-reforms-in-mecca/ (most recently accessed on 13 July 2020).

So far, the #MosqueMeToo movement has not gained much traction among Dutch Muslims. When asked, only a few women confirmed having heard about it. Moreover, except for one other woman besides Fatima, none of the women mentioned sexual harassment in their hajj stories. It cannot be excluded that the topic was not important to them or that none of them had suffered from it. Neither can it be excluded, however, that the women's specific situatedness at the intersection of different power structures has a 'muting effect' on them (cf. Ardener 1975). When explicitly asked about experiences of harassment, several of our interlocutors said that they had heard stories, but they would not elaborate or stated that they did not know what to think about such rumors. A few women reported having been touched or pinched in their buttocks during the *tawaf* themselves, while others explained that they had followed the advice of friends to wear protruding rucksacks or have their husband walk behind them. That they had apparently been given such advice indicates that what could not (yet) be said in the interview, could (already) be shared in more intimate circles of friends. As Fricker points out, breaking the silence to talk about injustices with friends in private settings or on social media platforms is often the starting point for subordinated groups to challenge their marginalization in dominant collective stories more widely (cf. Fricker 2007). As yet, considering the unfamiliarity of the majority of our interlocutors with the #MosqueMeToo movement, within the Dutch context female pilgrims apparently consider discussing harassment in private a more appropriate or effective option than going public by testifying on the internet.

## 5. In Conclusion: Reflections on the Dialogical Approach to Hajj Storytelling in a European Context

The pilgrimage to Mecca derives much of its symbolic power in the imagination of Muslims from the fact that Islam originated in Mecca. Stepping in the footsteps of the prophet Muhammad and other important religious role models to re-enact foundational episodes of Islam's historiography provides pilgrims with a sense of being in direct touch with this cherished past. Although the rites that make up the pilgrimage highlight continuity, the hajj stories discussed here demonstrate that these ritual re-enactments are not static, nor are the meanings that pilgrims attribute to them monolithic. Besides this interplay between continuity and change, another factor that makes hajj storytelling a particularly interesting research topic is that the flow of Muslims circling the Ka'ba is the symbol par excellence of the *umma*, the global community of Muslims. Stories about leaving one's daily lifeworld to assemble in Mecca with fellow pilgrims from all over the world provides insights in how local and global influences interweave in pilgrims' narrative construction of the meanings of the hajj in their lives. The stories discussed here exemplify that within the constraints of the specific power configurations in which they are embedded and the limitations of the discursive traditions available to them, Muslims appropriate the Islamic heritage in dialogue with prevailing normative interpretations of that heritage in ways that make it meaningful in their own lifeworld.

Adopting a dialogical approach in this article to hajj storytelling has enabled me to flesh out how the women whose stories were discussed attribute meanings to their pilgrimage experiences through ongoing dialogues with both collective and personal voices that address them in their daily lives. Contrary to popular views in Dutch society in which its Muslim citizens tend to be reduced to their 'Muslimness', this dialogical approach directs our attention to how being a Muslim intersects with other identifications in meaning making and finding one's place in the world. In this case study, applying a dialogical approach highlights specifically how the experiences of being a Muslim who has performed the pilgrimage to Mecca comes in the modality of being a gendered Muslim of a certain age group and having specific ethnic and national affiliations.

The main outcome of my analysis of the interviews is that the narrators position themselves in their hajj stories first and foremost as Dutch Muslims with Moroccan backgrounds. They do so mostly by distancing themselves from certain attitudes and practices concerning the pilgrimage to Mecca of the generation of their migrant parents, and by adopting ritual modes and meanings that are more in tune with their own quotidian experiences as Moroccan-Dutch citizens who live and grew up in the

Netherlands. More specifically, I have demonstrated that in formulating their own understandings of the meanings of the pilgrimage, the women creatively appropriate and mix vocabularies from a liberal-secular ethical tradition and a modern reformist strand within the Islamic ethical tradition.

Reading the women's hajj stories through the lens of dialogical positioning also reveals the limitations of the freedom available to individuals to adapt the vocabularies of discursive traditions to their own understanding. Such efforts are constrained by the meanings that these vocabularies have acquired within a specific configuration of power structures in which they were developed. This is specifically the case for those in subordinated positions, whose experiences tend to be marginalized in the regulative ideologies available to them to make sense of their lives (cf. Fricker 2007). For the narratives discussed here, this comes to the fore in the predicament female pilgrims find themselves in when wishing to address gender inequality in their hajj stories; physical harassment and gender discrimination have no place in conventional hajj storytelling in which the women insert their own stories. They do, however, feature widely in the popular negative representations of Islam in non-Muslim Dutch circles that the women do not wish to feed.

Related to the muting effect that regulative ideologies may have on the experiences of subordinated groups, the last issue that I wish to dwell on concerning the dialogical approach to hajj storytelling is my own position as researcher in the dialogical production of hajj stories in the interviews. The dilemma that Moroccan-Dutch female pilgrims face when considering whether or how to voice experiences of gender discrimination and harassment resonates with the epistemological 'impasse' Nadia Fadil's discusses in a review article about the state of the art in the anthropological study of Islam in Europe (Fadil 2019). Fadil argues convincingly that the anthropology of Islam in Europe is caught in an epistemological impasse. This deadlock is a consequence of the historical Orientalist discourse that continues to inform the dominant popular European frame in which Muslim citizens are presented as the abject 'Other'. One of the tensions caused by this frame revolves around the challenge to deconstruct representations of a binary opposition between a Western 'us' and a Muslim 'other', while simultaneously acknowledging the specificity of the religious experiences of European Muslim citizens. For anthropological scholarship on Islam in Europe this is reflected in the double bind researchers are confronted with when seeking to "account for the distinctiveness of ethical subjectivity of Muslims, while at the same time downplaying it"(Fadil 2019, p. 118).

In my own research, I have tried to deal with this paradox by alternating projects in which Islam is foregrounded with projects in which Islam is only a background presence. In the current hajj project, Islam obviously takes center stage. Regardless whether one studies Islam as a foreground or background presence, however, being affected by a discourse in which integration into European society is the yardstick that demarcates the distinction between the 'good' Muslim and the 'bad' Muslim cannot be avoided. The main argument in this article about the entanglement of different ethical discourses in the women's hajj stories, for example, might easily be read in terms of the 'domestication' of Islam in Dutch society in the unintended sense of Muslims being 'tamed' to fit the Dutch mold rather than in terms of a process of rooting and cultivation.

On a more concrete level, the inescapability of the double bind addressed by Fadil comes to the fore in the research design for this study. Opting for a topical, yet very open interview format was motivated by the objective to create space for our interlocutors to 'give voice' to their experiences in their own words.[27] As argued above, however, the dialogical nature of storytelling extends beyond the interview setting to include wider imagined audiences. This situation informs the sensitivities of both the interviewer and her interlocutors about what topics can be addressed easily and which ones are more delicate or best avoided altogether.

---

[27] Considering that it is the researcher who designs the interview-format and selects excerpts from the interviews to quote or analytically comment on, the realization of the ideal of 'giving voice' can always only be partial unless one works with only a very small number of research participants with whom one publishes together.

Gender discrimination is a case in point. The topics I wished to discuss in the interview were put in an order that allowed space to wait and see if women would address gender discrimination themselves. In case they did not, it was only brought up towards the latter part (but not at the end) of the interview.[28] This was motivated by the aim to create an atmosphere where our interlocutors might feel comfortable enough to also address more negative aspects of their experiences after having shared those they enjoyed looking back on or felt neutral about. In practice, however, when having to raise the issue as interviewer I noted a certain unease in myself. Additionally, it felt inappropriate to pursue the topic when I sensed that my interlocutors were hesitant to elaborate. In retrospect it is difficult to disentangle to what extent the reluctance to probe further was prompted by discerning discomfort that my interlocutors actually communicated verbally or through their body language, and to what extent I projected such discomfort onto them, maybe also out of fear that the women might conclude that my aim was to disclose Islam as an oppressive religion after all. I would therefore argue that the structural condition that accounts for the first dimension of the 'epistemological impasse' in the knowledge production on Islam in Europe identified by Fadil entraps both researchers and research participants.[29]

A second, interrelated dimension of the epistemological impasse discussed by Fadil concerns a "disciplinary nervousness" she notes among anthropologists about the epistemological claim making of the Islamic tradition (Fadil 2019, p. 121). Fadil's critique pertains first of all to the difficulties she observes in anthropological research on Islam in Europe to account for the constitutive weight of the Islamic tradition itself in the developments studied, and, secondly, to the already mentioned tendency to get trapped in an Euro-centric approach in which such developments are measured in terms of (in)compatibility with dominant models of modern European models of citizenship.

The stories about the pilgrimage to Mecca discussed here illustrate that, indeed, Europe is only one of several contexts that inspire European Muslims' religious lives; normative interpretations of stories from the Islamic tradition about the events and role models that are commemorated during the hajj and encounters with Muslims from other parts of the world also inform the religious views and practices of the women who feature in this article. At the same time, however, adopting the dialogical approach also reminds us that Fadil's proposition that an anthropology of Islam in Europe "ought to examine whether and how Islam is being (re)established as an autonomous tradition in a post-Christian Europe" (Fadil 2019, p. 127) cannot be realized. Discursive traditions do not operate fully autonomous. They are developed in dialogue with other discursive traditions by the people who inhabit them. Even ignoring or opposing other discourses implies active positioning. As Hafez (2011) and Schielke (2015) convincingly demonstrate for Egypt, for instance, through Muslims' engagement with a multiplicity of other collective voices that inform their lifeworlds, Islamic discourses and modes of reasoning get entangled with those of other discursive traditions in ways that go beyond dichotomies that pit the secular and modern against the religious and traditional. In terms of the impact of the historical marking of Islam as Europe's Other that Fadil reflects on in her review article, the biggest challenge in my own research on modern articulations of the pilgrimage to Mecca is to unpack the various discursive strands in these articulations without inadvertently suggesting that some are inherently European and others inherently Islamic. My not always equally successful attempts illustrate how difficult it is to adapt the vocabularies of the discursive traditions available to us to our own purposes.

---

[28] As researchers we have the responsibility to make the interview a rewarding experience for our research participants. Avoiding to address sensitive issues towards the conclusion of the interview is therefore motivated by the consideration that ending on a negative note might leave interlocutors with an uncomfortable feeling.

[29] A further complication is that people who are tired of having to prove that they are a 'good' Muslim and/or who fear that they will be portrayed as a 'bad' one whatever they say, tend to be disinclined to cooperate. As a consequence, a process of self-selection takes place, resulting in Muslims who are already motivated to discuss commonalities and differences between Muslim and non-Muslim European citizens being over-represented in interview projects such as the one on which this article is based. As the research projects of, for instance, the anthropologist Martijn de Koning among Salafi inclined Muslims illustrate, long term participant observation can alleviate this problem to some extent (cf. de Koning 2012, 2013).

**Funding:** This research was funded by the Netherlands Organization for Scientific Research (NWO), grant number: 360-25-150.

**Conflicts of Interest:** The author declare no conflict of interest.

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
