# Peer review of "Rearticulating the Conventions of Hajj Storytelling: Second Generation Moroccan-Dutch Female Pilgrims’ Multi-Voiced Narratives about the Pilgrimage to Mecca"

_religions, doi:10.3390/rel11070373_

Round 1
Reviewer 1 Report
It is quite remarkable that studies about hajj experiences, given their importance in the religious lives and theologies of Islam, are a relatively small section of the academic corpus on Islam in Europe. This article therefore is a welcome contribution to the field and quite obviously from someone who has worked on the issue in a very thorough and committed way. What is very important is how the author shows that her/his interlocutors’ stories are partly shaped and informed by prevailing discourses while also being ‘co-authors’ of the stories. (This by the way, begs the question if we as academics should include our interlocutors as authors of an article.)
The article overall is good if not excellent. I would suggest a few minor things, not as a necessary improvement but trying to inspire the author to push his/her ideas a little further.
1) in the introduction the author states “By adopting a ‘dialogical approach’ to self-storytelling, in this article I will discuss the interplay between content, narrator and lifeworld in the narrative construction of the meanings of the hajj.” What remains implicit however, is what a dialogical approach teaches us here?
2) The author, like many of us to be honest, seems to be caught up in what Fadil (2019, p. 122) has called a ‘double impasse’ in the (anthropological) study of Islam in Europe: "the Orientalist demarcation of Muslims as Other and the difficulty in representing Islam as a complex tradition, have fundamentally shaped scholarship on Islam in Europe…” For example, when the author in the introduction states: “A second reason which makes this particular category of pilgrims interesting to study is that the habitus of pilgrims who grew up in the Netherlands is informed by various discursive traditions simultaneously. I will argue that the expectations, experiences and the meanings the women whose stories are discussed attribute to the pilgrimage are the result of new sensibilities that they have acquired on the basis of how their positions in various cultural contexts intersect. A third reason why I concentrate on the hajj stories of women is that conceptions about female Muslim personhood constitute the site par excellence where the public debate about the (in)compatibility of being Muslim and being European or Dutch is played out.” Not contesting those two points at all, but it would be interesting to see the author engage with Fadil’s point in order to see if and how such an impasse indeed shapes his/her work and what the consequences might be. In particular because at some points in the article the author seems to repeat a dichotomy between ‘Islamic tradition’ and ‘Dutch cultural context’. For example in the third page, when she/he states: ‘Besides being informed by the Islamic tradition as transmitted to them by their parents, the habitus and the sense of self of women who grew up in the Netherlands are also shaped by the modern liberal discourse and the culture of consumerism that dominate in the Dutch cultural contexts in which they were educated, earn their living, and lead their social lives.’. First of all, the transmission of the Islamic tradition is done by their parents in the Dutch context so already shaped by it (and by the experience of migration) and one could question whether the role of the parents is indeed that strong given the influence of the internet for example. Second, by stating ‘also shaped by’ the author (unintended I think) slips into the idea of seeing ‘modern liberal discourse and the culture of consumerism’ as a tradition or context that can easily be distinguished from Islamic tradition while that is definitely not the point that is being made (and therefore bringing us back to Fadil’s idea of ‘impasse’).
3) I was struck by the idea of a ‘loving’ and ‘nurturing’ God. This idea, or rather the emphasis on it, seems to make the religiosity of the author’s interlocutors distinct from many categorized as Salafists and/or jihadists. Who do know and profess this idea, yet also push and declare the idea of a vengeful, powerful God who strikes when deemed necessary. Is such an idea completely absent in the narratives of the interlocutors?
4) The author discusses the discourse of authenticity as one among other others informing the religiosity of those for whom obedience is not enough (p. 8, line 384 – 387 for example). It is an idea of authenticity which can be based upon readings of the Islamic tradition as well as one which has gained a lot of traction in different religious and other traditions (see the work by Heelas, Taylor). Does this mean this idea of authenticity can only be found among the younger generation or is the idea that one ‘simply’ has to be obedient, a different kind of authenticity. And how does sincerity play a role here as there is an interesting and complicated relation between authenticity and sincerity (Trilling, 2009).
5) The reference to #MosqueMeToo deserves a little bit of explanation here I think. And did this movement gain any traction in the Dutch context?
Fadil, N. (2019). The Anthropology of Islam in Europe: A Double Epistemological Impasse. Annual Review of Anthropology, 48(1), 117-132. doi:10.1146/annurev-anthro-102218-011353
Trilling, L. (2009). Sincerity and authenticity: Harvard University Press.
Reviewer 2 Report
This was an excellent paper to read--I wanted to read more about this topic! I think that this paper is timely and really helps bring women's perspectives to the fore regarding identity and pilgrimage. I appreciated the use of theoretical concepts and ideas to help me as a reader to better understand the story-telling of these women.
I recommend this paper be accepted with minor revisions:
I was not entirely clear initially on the purpose of this paper. Maybe make this a bit clearer at the beginning?
The use of I versus we/our in this paper--I am assuming that this is a single-authored paper?
ln 63: "two categories of people whose"
ln 67: "are discussed attribute to the pilgrimage are: - awkward phrasing
ln 75 "pilgrim", not "pilgrims"
ln 89: What is this work by Mikhail Bakhtin that the authors builds upon? unclear
ln 97: What is meant by 'ideal stories'?
ln 333: What is the translation of the Dutch proverb?
The conclusion needs work. Yes, the author demonstrated this and that, but where is the so what? What is the next step in the research? Why should people care about what the author wrote? I am left wanting to see the value of this, even though the value is in the text already. The author needs to state this clearly to their audience.
